 | Microbiology **Spectrum**

ⓐ | **Open Peer Review** | Environmental Microbiology | Observation

# Soil microbiome analysis supports claims of ineffectiveness of *Pseudomonas fluorescens* D7 as a biocontrol agent of *Bromus tectorum*

Gordon F. Custer,[1,2,3,4,5] Brian A. Mealor,[6,7,8] Beth Fowers,[6,7,8] Linda T. A. van Diepen[1,2,8]

**ABSTRACT** Cheatgrass (*Bromus tectorum*) is one of the most problematic invasive plants in the western United States. Invasion by annual grasses disrupts nutrient cycling and negatively affects above- and below-ground biodiversity. Land managers use chemical herbicides, mechanical controls, cultural practices, and bioherbicides to combat this invasive plant. Recently, the biocontrol agent *Pseudomonas fluorescens* D7 has been touted as a non-chemical herbicide that offers lasting control of cheatgrass. However, experimental results report limited effectiveness under field conditions. To understand the underlying cause of the variable efficacy of this commercially available bioherbicide, soil samples from an active cheatgrass invasion treated with *P. fluorescens* D7 were collected and screened using high-throughput sequencing. At 18 months post-application, the bioherbicide had limited lasting effects on bacterial community composition, and few reads assigned to *P. fluorescens* were found in our sequence data. We suggest that the failure to establish sufficiently may partially explain the inability of this biocontrol agent to suppress *B. tectorum* under field conditions.

**IMPORTANCE** Cheatgrass is one of North America's most problematic invasive species. Invasion by this annual grass alters ecosystem structure and function and has proven very challenging to remove with traditional approaches. Commercially available bioherbicides, like *P. fluorescens* D7, are applied with the goal of providing lasting control from a single application. However, experimental results suggest that this bioherbicide has limited efficacy under field conditions. Potential explanations for variable efficacy include a failure of this bioherbicide to establish in the soil microbiome. However, to our knowledge, no data exist to support or refute this hypothesis. Here, we use a deep-sequencing approach to better understand the effects of this bioherbicide on the soil microbiome and screen for *P. fluorescens* D7 at 18 months post-application.

**KEYWORDS** cheatgrass, biological invasion, herbicide, invasive plants, bioherbicide

Invasive plants, like cheatgrass (*Bromus tectorum*), are problematic across the western United States and pose a threat to ecosystem function, structure, and stability. Following invasion, cheatgrass disrupts carbon and nitrogen cycling as well as the delicate balance of many plant-microbe interactions (1–3). Managing this problematic invader is a top concern for land managers, as invaded lands are often less suitable for grazing and can lead to decreased biodiversity across trophic levels (4) and an increased frequency of wildfires (5).

Approaches to control populations of cheatgrass include chemical herbicides, targeted grazing, mechanical removal, and more recently the application of bioherbicides. One such commercially available bioherbicide, *Pseudomonas fluorescens* D7, has been touted as a non-chemical alternative for interannual cheatgrass control, and this

Address correspondence to Linda T. A. van Diepen, linda.vandiepen@uwyo.edu.

See the funding table on p. 4.

The authors declare no conflict of interest.

*[This article was published on 5 December 2023 with errors in the title, abstract, text, and supplemental material. These were corrected in the current version, posted on 18 December 2023.]*

reportedly is achieved through a single application. This bioherbicide is reported to have no negative effects on the native plant community, and these claims are supported by early greenhouse experiments (6, 7). However, more recently, a large-scale synthesis from the inter-Mountain West reported "little evidence of effectiveness" (8–10) and calls into question the use of this commercial inoculant. A potential explanation for the limited efficacy of this bioherbicide is that *P. fluorescens* D7 fails to establish in soils and thus cannot suppress the invasive annual grass over subsequent growing seasons. However, to date, there is little information to support or refute this working hypothesis (9). To address this knowledge gap, we set out to examine (i) whether a single application of *P. fluorescens* D7 is sufficient to establish a detectable population in soils, and (ii) whether application of this bioherbicide has lasting effects on soil microbial diversity at 18 months post-application.

We used a commercial inoculum from Verdesian Life Sciences (Cary NC, UA) reconstituted at $2 \times 10^{11}$ *P. fluorescens* D7 cells/0.1 g. The bioherbicide was applied to cheatgrass-invaded plots near Buffalo, WY, USA in November 2015 at 0.0, 0.49, and 4.94 g/ha (0×, 1×, and 10× recommended application rates, respectively) using an all-terrain-vehicle with a mounted "boomless" sprayer. Experimental plots were also treated with the chemical herbicide imazapic, applied at two different rates (52.6 or 87.7 g ai/ha) at this time. This represents a common approach of land managers to apply multiple control measures simultaneously to achieve cheatgrass control (10). Treatments were applied in November as winter snows increased the likelihood of elevated soil moisture and enhanced bioherbicide establishment. Soil samples were collected in May 2017 (18 months post-application) and prepared for 16S rRNA meta-barcoded amplicon sequencing of the V4 region using the 515f and 806r primer pair on an Illumina MiSeq platform. Sequence data were processed using DADA2 v1.16 (11), cutadapt (12), and Phyloseq (13) in R V4.1.1. The effect of bioherbicide application on bacterial α- and β- diversity was assessed using analysis of variance (ANOVA) and permutational multivariate analysis of variance (PERMANOVA) (14). Next, to screen for populations of *P. fluorescens* D7 in our soil microbiome data set, all Amplicon Sequence Variants (ASVs) assigned as members of the family Pseudomonadaceae by Silva database V138 were checked using BLASTN. The top 5,000 results were screened for "*fluorescens*" and "D7." For a full description of the experimental design and methods, please refer to the supplementary materials.

To answer our question of whether a single application of *P. fluorescens* D7 under field conditions can lead to the establishment of interannual populations, we screened our sequence data set for ASVs assigned to the family Pseudomonadaceae. We found that they accounted for a very small percentage of the total reads (<0.01% in all cases). Kruskal-Wallis testing showed no statistical differences in the relative abundance of these three members of Pseudomonadaceae across the bio-pesticide application rates and controls ($P > 0.05$). We report that two ASVs of the family Pseudomonadaceae have potential BLASTN alignments matching *P. fluorescens* (Table S1). However, neither returned a match to strain "D7" in the first 5,000 BLASTN hits, though this may be due to the strain-level sequence not being represented in the BLASTN database. Please see supplementary materials for a full report of our results. To determine whether the application of bioherbicide has a lasting effect on bacterial community composition, we used PERMANOVA testing and reported no significant effect of application rate on bacterial β-diversity ($P = 0.19$, $F_{2,22} = 1.722$; Fig. S2). These results suggest that a single application of the surface spray inoculation of *P. fluorescens* D7 did not result in an abundant population in soil and may partially explain its limited efficacy under field conditions. It is worth noting, however, that the application of imazapic could impede the establishment of *P. fluorescens* D7 by killing the cheatgrass hosts. Despite this, we still expected to find higher abundances of the bioherbicide taxon in the soil environment of the inoculated plots compared to the controls, yet this was not the case (Table S2). To our surprise, the application of *P. fluorescens* D7 at 4.94 g/ha resulted in a statistically significant decrease in bacterial richness at 18 months as compared to the 0.49 g/ha

treatment (Fig. S3; 4.94 g/ha – 450 ± 53, 0.49 g/ha – 526 ± 69; $P < 0.05$, $F_{2,22} = 4.517$). However, neither was statistically different from the untreated controls.

Although there was no discernable difference in the abundance of potential members of *P. fluorescens* between the application rates and controls at 18 months post-treatment, application at higher rates may result in a lasting reduction in local soil bacterial diversity and potentially prime the system for subsequent invasion by microbes through additional amendments (15). Previous experimental work has shown that transient microbial invaders (i.e., those that fail to establish persistent populations) can induce shifts in ecosystem structure and function and result in alternate stable states (15, 16). For example, even when failing to establish in experimental soil microcosms, a microbial invader may steer native ecosystem function away from their niche. In this case, the native microbial community's carbon use profile (determined by Biolog plates) shifts away from the invader's due to competitive interactions (15). This shift can lead to a legacy effect where open niche space may increase the probability of successful invasion during future introductions (15). Interpreting this previous work in line with our findings, multiple applications of this bioherbicide may be necessary to facilitate the long-term establishment and promotion of interannual cheatgrass control (17). Additionally, future work should include positive controls of inoculant viability and intermediate samplings to determine the temporal dynamics of inoculant establishment over shorter time frames. This information can assist with improving inoculation techniques.

Akin to the processes that lead to the establishment of cheatgrass and other invasive species, *P. fluorescens* D7 must successfully pass through the stages and barriers of biological invasion to become a persistent member of the soil microbiome (18). Leveraging ecological processes like dispersal may be one way to improve outcomes in these complex systems (17). For example, increasing propagule pressure through multiple independent introductions or introducing more cells at each event may lead to an increase in establishment rate, although we did not observe this across our application rates. Given the considerable interest in biological amendments for agricultural and rangeland management, understanding the underlying ecology and improving inoculation and establishment methods are valuable avenues of future research (17, 19–21). Methods that spray a bacterial solution onto the soil surface assume that it is sufficient to promote establishment. However, this method leaves bacteria susceptible to unfavorable conditions like high temperatures or desiccation (10). Furthermore, they ignore the ecology of these systems and the fact that the microorganism must invade the existing soil microbial community, overcome biotic resistance, and outcompete native taxa (17, 22). In conclusion, our results show that even at 10 times the recommended application rates, abundant populations of *P. fluorescens* were not recovered in our soil samples, and this may partially explain the limited efficacy of this bioherbicide under field conditions. Not only do microbial communities often work as a cohort but microbial functions that suppress weeds are often affected by both microbial interactions and environmental conditions that can change in their non-native habitats. With this, our results indicate that improved inoculation methods are needed (23–25) to facilitate establishment before product efficacy can be rigorously assessed.

## AUTHOR AFFILIATIONS

[1]Department of Ecosystem Science and Management, University of Wyoming, Laramie, Wyoming, USA

[2]Program in Ecology, University of Wyoming, Laramie, Wyoming, USA

[3]Department of Plant Science, The Pennsylvania State University, University Park, Pennsylvania, USA

[4]Huck Institutes of the Life Sciences, The Pennsylvania State University, University Park, Pennsylvania, USA

[5]The One Health Microbiome Center, The Pennsylvania State University, University Park, Pennsylvania, USA

[6]Department of Plant Sciences, University of Wyoming, Laramie, Wyoming, USA

[7]Sheridan Research and Extension Center, Sheridan, Wyoming, USA

[8]Institute for Managing Annual Grasses Invading Natural Ecosystems, Sheridan, Wyoming, USA

## AUTHOR ORCIDs

Gordon F. Custer http://orcid.org/0000-0002-4328-0714

Brian A. Mealor http://orcid.org/0000-0002-8923-342X

Linda T. A. van Diepen http://orcid.org/0000-0002-5902-9364

## FUNDING

| Funder | Grant(s) | Author(s) |
|---|---|---|
| U.S. Department of Agriculture (USDA) | 1002941 | Brian A. Mealor |
| National Science Foundation (NSF) | EPS-1655726 | Linda T. A. van Diepen |

## AUTHOR CONTRIBUTIONS

Gordon F. Custer, Formal analysis, Software, Validation, Writing – original draft, Writing – review and editing | Brian A. Mealor, Conceptualization, Funding acquisition, Investigation, Methodology, Project administration, Resources, Supervision, Writing – review and editing | Beth Fowers, Investigation, Methodology, Resources, Writing – review and editing | Linda T. A. van Diepen, Formal analysis, Funding acquisition, Project administration, Writing – original draft, Writing – review and editing

## DATA AVAILABILITY

Raw sequence data can be found under NCBI SRA project number PRJNA962488.

## ADDITIONAL FILES

The following material is available online.

### Supplemental Material

**Supplemental file (spectrum01771-23Supp_Materials_Custer_et_al_2023.docx).** Detailed methods and results.

### Open Peer Review

**PEER REVIEW HISTORY (review-history.pdf).** An accounting of the reviewer comments and feedback.

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
