## [Reviewer comments · Microbiology Spectrum]

Microbiology Spectrum

Failure to recover *Pseudomonas fluorescens* D7 supports claims of ineffectiveness as biocontrol agent of *Bromus tectorum*

Gordon Custer, Brian Mealor, Beth Fowers, and Linda van Diepen

Corresponding Author(s): Gordon Custer, The Pennsylvania State University

Review Timeline:

Submission Date:	April 28, 2023
Editorial Decision:	May 6, 2023
Revision Received:	July 24, 2023
Editorial Decision:	August 24, 2023
Revision Received:	October 24, 2023
Accepted:	October 29, 2023

Editor: Frédérique Reverchon

Reviewer(s): Disclosure of reviewer identity is with reference to reviewer comments included in decision letter(s). The following individuals involved in review of your submission have agreed to reveal their identity: Alana Den Breeyen (Reviewer #1); Kurt Reinhart (Reviewer #2)

Transaction Report:

DOI: <https://doi.org/10.1128/spectrum.01771-23>

May 6, 2023

Dr. Gordon Fritz Custer
The Pennsylvania State University
University Park, PA 16802

Re: Spectrum01771-23 (Failure to recover *Pseudomonas fluorescens* D7 supports claims of ineffectiveness as biocontrol agent of *Bromus tectorum*)

Dear Dr. Gordon Fritz Custer:

The manuscript is short and I have read it myself, however there is a total lack of description of methods, replicate, controls, which really hinders the capacity to assess result validity.

I consulted with the Spectrum staff and they recommended that you may submit your manuscript as a Observation article. Should you wish to proceed with a New data letter format, I would recommend that you include a proper method description and result report, at least in Supplementary Materials.

The Spectrum staff will be contacting you shortly to guide you through the resubmission process.

Link Not Available

Sincerely,

Frédérique Reverchon

Journals Department
Reviewer comments:

Staff Comments:

Preparing Revision Guidelines

To submit your modified manuscript, log onto the eJP submission site at <https://spectrum.msubmit.net/cgi-bin/main.plex>. Go to

Author Tasks and click the appropriate manuscript title to begin the revision process. The information that you entered when you first submitted the paper will be displayed. Please update the information as necessary. Here are a few examples of required updates that authors must address:

Please return the manuscript within 60 days; if you cannot complete the modification within this time period, please contact me. If you do not wish to modify the manuscript and prefer to submit it to another journal, please notify me of your decision immediately so that the manuscript may be formally withdrawn from consideration by Microbiology Spectrum.

Reviewer comments:

The previous version of this manuscript was not sent for peer-review. As such, there are no reviewer comments to address.

Editor comments:

The manuscript is short and I have read it myself, however there is a total lack of description of methods, replicate, controls, which really hinders the capacity to assess result validity.

I consulted with the Spectrum staff and they recommended that you may submit your manuscript as a Observation article.

Should you wish to proceed with a New data letter format, I would recommend that you include a proper method description and result report, at least in Supplementary Materials.

The manuscript has been reformatted in line with the guidelines for an Observation.

August 24, 2023

Dr. Gordon Fritz Custer
The Pennsylvania State University
University Park, PA 16802

Re: Spectrum01771-23R1 (Failure to recover *Pseudomonas fluorescens* D7 supports claims of ineffectiveness as biocontrol agent of *Bromus tectorum*)

Dear Dr. Gordon Fritz Custer:

I have now received the comments made by two independent reviewers on your manuscript. They both recommend some modifications, as you will see below. As your manuscript has been submitted as an "observation", I acknowledged that the amount of additional experiments or in depth discussion should be limited. However, there are some methodological choices that should be described more thoroughly, as pointed by Reviewer #2.

Link Not Available

Sincerely,

Frédérique Reverchon

Journals Department
Reviewer comments:

Reviewer #1 (Comments for the Author):

I have made comments and suggestions directly on the attached merged file.
It would have been interesting to see if you could have isolated the bacteria from the soil to confirm your sequencing results.

Reviewer #2 (Comments for the Author):

Cheatgrass invasion is a problem throughout the western USA. The bacterial D7 strain was thought to be a miracle cure for cheatgrass. The hype has waned since independent groups started publishing results showing it failed to affect cheatgrass. Now many are interested in understanding when/where does/doesn't D7 control cheatgrass to better understand where to use it? A central question is whether D7 is even establishing?

Gordon et al. used molecular techniques to survey/check for D7 18 months after field applications and whether it may have disrupted microbial/bacterial community composition. I appreciate that a group is using molecular techniques to measure the presence and abundance of D7 and other microbes. I've heard of at least three groups interested in using molecular techniques to monitor whether D7 establishes (or not) in areas invaded by cheatgrass.

I appreciate that this group is the first to get something out on this topic. There are aspects of the study that are less than desirable like 1) lack of a positive control(s), 2) all field plots with D7 applications were also treated with herbicide, & 3) lack of caveat statements. Assuming treated plots controlled cheatgrass then I think your study helps lay to rest the notion that D7 needs to be applied with herbicide as it seemingly failed to establish and effects were likely 100% from the herbicide. Companies did some great marketing of off label use.

At the very least, the authors need some major revision to the discussion to critically report strengths and weaknesses of their study and how others should proceed. Readers will appreciate that the group sampled what was available and not necessarily what would have been ideal in hindsight.

Additional info-

Ann Kennedy did some early work with antibiotic resistant D7 & ACK55 strains. She used plate counts of soil dilutions to confirm short term D7 establishment (Johnson et al., 1993; Stubbs et al., 2014). Ann also reported longer term presence of ACK55 at field sites (Kennedy, 2018).

She wrote- "During a thaw in February following application in mid-fall, the bacterial strains were in soil at 5×10^8 cfu g⁻¹ soil. Two years after application, February bacterial populations in soil were at 6×10^7 cfu g⁻¹ soil. By the fourth year after application, the rifampicin-resistant populations in soil declined to levels of 3×10^5 cfu g⁻¹ soil, which was the lowest level that showed inhibition in the laboratory. Bacterial populations decreased faster in soils at sites with lower precipitation, although bacterial populations were not always different due to site moisture. Five to six years after application, the bacterial populations were below reliable detection levels in the field soil."

Additional comments-

L64 change "no" to "little". (& L124-6)

Note that Reinhart et al. (2019) performed Petri-plate, pot, and field experiments. The bioherbicide affected cheatgrass in Petri-plate but not in pots and field experiments thereby suggesting properties of regional soils (e.g. nutrients, texture, pH) may limit establishment of bioherbicide. Two pot experiments were also conducted that help to parse effects of resident microbes from other soil properties. For example, one experiment had pots filled with sterilized soils (main manuscript), two seed sources, etc. and another had varying soil types and soil inoculants (soil inoculant, soil inoculant+ACK55, ACK55; in supplement). These experiments were at ~optimal conditions (cool & wet) for bioherbicide, and they confirmed starting bacterial densities (so there was no question bacteria were originally alive and at a known density)...

L71 I'm sort of viewing your study as a trial and proof of concept. However, please include more details on experimental and sampling designs (e.g. plot #, samples per plot?). Or direct readers to the supplement for more detail.

L71 Good treatments & well timed. Ann Kennedy would recommend applying D7 prior to rain/snow events so that the water would help carry the bacteria into the soil. Was this likely? How many plots? Not sure the experiment design was fully described even in the supplement (e.g. total number of plots, subplots?).

L74 Is this a good region for bacterial community composition? I assume yes but citations might be nice.

L80-2 Is this an informative region capable of differentiating species and isolates (for lack of a better word) and was there a positive control? Other evidence that this can A) detect *P. fluorescens* in general and B) differentiate D7 from other genotypes? Ann indicated the molecular tools (Ibekwe et al., 2010) were not able to detect D7 was that wrong?

L88 please list actual P-values throughout.

L96 This could be a sign of two possible legacy effects: effect of nutrients in carrier solution (assuming residual growth media varied with concentration differences) or effect of D7 (though it seems unlikely since they were evidently more abundant 18 months later).

General comment- Microbial community descriptions may describe a large proportion of non-active soil microbes. Is it possible D7 was more prevalent during the winter months? I think this is unlikely for WY but it was isolated from soils in eastern

Washington which likely has more favorable conditions for activity in winter (moist soils & not too cold) and possibly dramatically different soils (e.g. pH) than WY... Ann did a lot of her monitoring of Ack55 in winter.

L129 Comment- I suspect we need to have a better understanding of where the soil and abiotic conditions are suitable for D7 establishment relative to where D7 was locally adapted.

Discussion-

The discussion is a bit too long for such thin results. I would cut some of the speculative discussion content and add caveat content and future directions (or best practices). For example, it would have been nice to have samples collected pre-treatment, 1 wk post-treatment, & 18 months post-treatment (and in areas without herbicide). A positive control(s) would have been reassuring that the magic freeze-dried stuff was actually viable and D7. Having D7 applications with herbicide may have contributed to failure of D7 to establish. The isolate is thought to feed off of cheatgrass root exudates over winter. If you kill cheatgrass seedlings and few/none are overwintering then you likely starved D7. I suppose there is also the issue of DNA techniques describing numerous biota that are not active versus active. Many questions.

Supplement-

L31 Is it possible that the herbicide confounded detecting D7? My impression from Ann is that the bacteria should proliferate with cheatgrass and feed off of its root exudates (i.e. is closely associated with its roots but doesn't cause noticeable pathology of roots). If cheatgrass was suppressed by herbicide then that might hamstring detecting D7. To me, this herbicide + bioherbicide application scheme was created out of some business scheme and not biological processes. Ann gave me numerous documents related to instruction on use and none that I recall suggested applying D7 with herbicide. That would ignore the life cycle of D7.

L32 not sure about "synergistic effects" but realize vendor reps were pushing this idea at the time...

L33 Not seeing the citation for #1.

L57-8 You have a negative control. However, I wish that you had a positive control or something showing that the bacteria were ever alive and capable of establishing. I have questions about whether the inoculant was ever viable, had D7 to begin with, whether something about the application killed D7.

L104 Have you done any trials to confirm that D7 is detectable in treated soils with other *Pseudomonas* &/or that the inoculant was ever viable and containing D7?

References

- Ibekwe, A. M., Kennedy, A. C., & Stubbs, T. L. (2010). An assessment of environmental conditions for control of downy brome by *Pseudomonas fluorescens* D7. *International Journal of Environmental Technology and Management*, 12(1), 27-46.
- Johnson, B. N., Kennedy, A. C., & Ogg, A. G. (1993). Suppression of downy brome growth by a rhizobacterium in controlled environments. *Soil Science Society of America Journal*, 57(1), 73-77.
<https://doi.org/10.2136/sssaj1993.03615995005700010014x>
- Kennedy, A. C. (2018). Selective soil bacteria to manage downy brome, jointed goatgrass, and medusahead and do no harm to other biota. *Biological Control*, 123, 18-27. <https://doi.org/https://doi.org/10.1016/j.biocontrol.2018.05.002>
- Reinhart, K. O., Carlson, C. H., Feris, K. P., Germino, M. J., Jandreau, C. J., Lazarus, B. E., Mangold, J., Pellatz, D. W., Ramsey, P., Rinella, M. J., & Valliant, M. (2019). Weed-suppressive bacteria fails to control *Bromus tectorum* under field conditions. *Rangeland Ecology & Management*, accepted 7/25/2019.
- Stubbs, T. L., Kennedy, A. C., & Skipper, H. D. (2014). Survival of a rifampicin-resistant *Pseudomonas fluorescens* strain in nine mollisols. *Applied and Environmental Soil Science*, 2014, 7, Article 306348. <https://doi.org/10.1155/2014/306348>

Staff Comments:

Preparing Revision Guidelines

- Point-by-point responses to the issues raised by the reviewers in a file named "Response to Reviewers," NOT IN YOUR COVER LETTER.

- Upload a compare copy of the manuscript (without figures) as a "Marked-Up Manuscript" file.
- Each figure must be uploaded as a separate file, and any multipanel figures must be assembled into one file.
- Manuscript: A .DOC version of the revised manuscript
- Figures: Editable, high-resolution, individual figure files are required at revision, TIFF or EPS files are preferred

Please return the manuscript within 60 days; if you cannot complete the modification within this time period, please contact me. If you do not wish to modify the manuscript and prefer to submit it to another journal, please notify me of your decision immediately so that the manuscript may be formally withdrawn from consideration by Microbiology Spectrum.

**Title: Failure to recover *Pseudomonas fluorescens* D7 supports claims of ineffectiveness as**
**biocontrol agent of *Bromus tectorum***

**Authors:** Gordon F. Custer^{a,b,c,d,e}, Brian A. Mealor^{f,g,h}, Beth Fowers^{f,g,h}, and Linda T.A. van
Diepen^{a,b,h*}

**Authors' affiliations:**

7 ^a Department of Ecosystem Science and Management, University of Wyoming, Laramie, WY,
USA

9 ^b Program in Ecology, University of Wyoming, Laramie, WY, USA

10 ^c Department of Plant Sciences, Pennsylvania State University, University Park, PA, USA

11 ^d Huck Institutes of the Life Sciences, The Pennsylvania State University, University Park, PA,
12 USA

13 ^e The One Health Microbiome Center, Huck Institutes of the Life Sciences, The Pennsylvania
14 State University, University Park, PA, USA

15 ^f Department of Plant Sciences, University of Wyoming, Laramie, WY, USA

16 ^g Sheridan Research and Extension Center, Sheridan, WY, USA

17 ^h Institute for Managing Annual Grasses Invading Natural Ecosystems, Sheridan, WY, USA

*Corresponding author: LvD – Email: linda.vandiepen@uwyo.edu

**Keywords:** cheatgrass, biological invasion, herbicide, invasive plants

**Abstract:** Cheatgrass (*Bromus tectorum*) is one of the most problematic invasive plants in the
western United States. Invasion by annual grasses disrupts nutrient cycling and negatively affects
above- and below-ground biodiversity. Land managers use chemical herbicides, mechanical
controls, cultural practices, and biopesticides to combat this invasive plant. Recently, the
biocontrol agent *Pseudomonas fluorescens* D7 has been touted as a non-chemical herbicide that
offers lasting control of cheatgrass. However, experimental results report limited effectiveness
under field conditions. To understand the underlying cause of the variable efficacy of this
commercially available biopesticide, soil samples from an active cheatgrass invasion treated with
*P. fluorescens* D7 were collected and screened using high-throughput sequencing. At 18 months
post-application, the biopesticide had limited lasting effects on bacterial community
composition, and no reads assigned to *P. fluorescens* D7 were found in our sequence data. We
suggest that the failure to establish may partially explain the inability of this biocontrol agent to
suppress *B. tectorum* under field conditions.

**Importance:** Cheatgrass is one of North America's most problematic invasive species. Invasion
by this annual grass alters ecosystem structure and function and has proven very challenging to
remove with traditional approaches. Commercially available biopesticides, like *P. fluorescens*
D7, are applied with the goal of providing lasting control from a single application. However,
experimental results suggest that this biopesticide has limited efficacy under field conditions.
Potential explanations for variable efficacy include a failure of this biopesticide to establish in
the soil microbiome. However, to our knowledge, no data exists to support or refute this
hypothesis. Here, we use a deep sequencing approach to better understand the effects of this
biopesticide on the soil microbiome and screen for *P. fluorescens* D7 at 18 months post-
application.

**Observation**

Invasive plants, like cheatgrass (*Bromus tectorum*), are problematic across the western United
States and pose a threat to ecosystem function, structure, and stability. Following invasion,
cheatgrass disrupts carbon and nitrogen cycling as well as the delicate balance of many plant-
microbe interactions (1–3). Managing this problematic invader is a top concern for land
managers, as invaded lands are often less suitable for grazing and can lead to decreased
biodiversity across trophic levels (4), and an increased frequency of wildfires (5).

Approaches to control populations of cheatgrass include the application of chemical
herbicides, targeted grazing, mechanical removal, and more recently the application of
biopesticides. One such commercially available biopesticide, *Pseudomonas fluorescens* D7, has
been touted as a non-chemical alternative for interannual cheatgrass control, and this reportedly
is achieved through a single application. This biopesticide is reported to have no negative effects
on the native plant community, and these claims are supported by early greenhouse
experimentation (6, 7). However, more recently, a large-scale synthesis from the inter-Mountain
West reported “little evidence of effectiveness” (8–10), and calls into question the use of this
commercial inoculant. A potential explanation for the limited efficacy of this biopesticide is that
*P. fluorescens* D7 fails to establish ~~itself~~ in soils, and thus cannot suppress the invasive annual
grass over subsequent growing seasons. However, to date, there is no information to support or
refute this working hypothesis. To address this gap in understanding, we set out to examine 1)
whether a single application of *P. fluorescens* D7 is sufficient to establish a detectable population
in soils, and 2) whether application of this biopesticide has lasting effects on soil microbial
diversity at 18 months post-application.

We used a commercial inoculum from Verdesian Life Sciences (Cary NC, UA)
reconstituted at 2×10^{11} *P. fluorescens* D7 cells / 0.1 g. The biopesticide was applied to
cheatgrass-invaded plots near Buffalo, WY, USA in November 2015 at 0.0 g/ha, 0.49 g/ha, and
4.94 g/ha (0x, 1x, and 10x recommended application rates, respectively) using an ATV with a

[revised manuscript text omitted]

- 14. Oksanen J, Blanchet FG, Friendly M, Kindt R, Legendre P, McGlenn D, Minchin PR, O’Hara
R, Gavin L, Simpson P, Solymos M, Stevens HH, Szoecs E, Wagner H. 2017. vegan:
Community Ecology Package. R package version 2.4-3.
- 15. Mallon CA, Le Roux X, van Doorn GS, Dini-Andreote F, Poly F, Salles JF. 2018. The impact
of failure: unsuccessful bacterial invasions steer the soil microbial community away from
the invader’s niche. *ISME J* 12:728–741.
- 16. Amor DR, Ratzke C, Gore J. 2020. Transient invaders can induce shifts between
alternative stable states of microbial communities. *Sci Adv* 6:eaay8676.
- 17. King WL, Bell TH. 2022. Can dispersal be leveraged to improve microbial inoculant
success? *Trends Biotechnol* 40:12–21.

- 18. Mallon CA, Elsas JD van, Salles JF. 2015. Microbial Invasions: The Process, Patterns, and
Mechanisms. *Trends Microbiol* 23:719–729.
- 19. Haskett TL, Tkacz A, Poole PS. 2020. Engineering rhizobacteria for sustainable agriculture.
*ISME J.* 15:949-964.
- 20. Albright MBN, Louca S, Winkler DE, Feeser KL, Haig S-J, Whiteson KL, Emerson JB, Dunbar
186 J. 2021. Solutions in microbiome engineering: prioritizing barriers to organism
establishment. *ISME J.* 16:331-338.
- 21. Wubs ERJ, Van Der Putten WH, Bosch M, Bezemer TM. 2016. Soil inoculation steers
restoration of terrestrial Ecosystems. *Nat Plants* 2:1–5.
- 22. Kaminsky LM, Trexler R V., Malik RJ, Hockett KL, Bell TH. 2019. The Inherent Conflicts in
Developing Soil Microbial Inoculants. *Trends Biotechnol* 37:140–151.
- 23. Van Dyke MI, Prosser JI. 2000. Enhanced survival of *Pseudomonas fluorescens* in soil
following establishment of inoculum in a sterile soil carrier. *Soil Biol Biochem* 32:1377–
1382.
- 24. Santos MS, Nogueira MA, Hungria M. 2019. Microbial inoculants: reviewing the past,
discussing the present and previewing an outstanding future for the use of beneficial
bacteria in agriculture. *AMB Express* 9.
- 25. Fukami J, Nogueira MA, Araujo RS, Hungria M. 2016. Accessing inoculation methods of
maize and wheat with *Azospirillum brasilense*. *AMB Express* 6:1–13.

Reviewer #1 (Comments for the Author):

I have made comments and suggestions directly on the attached merged file.

All suggestions made as comments have been incorporated in the revised manuscript.

It would have been interesting to see if you could have isolated the bacteria from the soil to confirm your sequencing results.

While we agree with this sentiment, we are unable to do so this long after application. With that, we have now noted this as a caveat to the interpretation of our results.

Interestingly it is suggested that you could apply the bioherbicide at 2g/acre up to four times in a 12 month period for a maximum application of 8g/acre.

See previous comment regarding number of applications. Why only test a single application instead of up to the maximum suggested amount? Previous trials also only apply once or twice within a two month period.

With respect to the number of independent applications, the product label has many suggestions. It would be infeasible to apply all the different combinations. We opted for a single application at two different rates to simplify the logistics of applying in rangelands and still acted within the label recommendations. The different application rates used align with the concept of propagule pressure (specifically propagule size) from invasion biology theory and thus have support from the broader literature. The rates used in this study also aligned with an ongoing in-field evaluation by land managers in the region.

Reviewer #2 (Comments for the Author):

Cheatgrass invasion is a problem throughout the western USA. The bacterial D7 strain was thought to be a miracle cure for cheatgrass. The hype has waned since independent groups started publishing results showing it failed to affect cheatgrass. Now many are interested in understanding when/where does/doesn't D7 control cheatgrass to better understand where to use it? A central question is whether D7 is even establishing?

Gordon et al. used molecular techniques to survey/check for D7 18 months after field applications and whether it may have disrupted microbial/bacterial community composition. I appreciate that a group is using molecular techniques to measure the presence and abundance of D7 and other microbes. I've heard of at least three groups interested in using molecular techniques to monitor whether D7 establishes (or not) in areas invaded by cheatgrass.

I appreciate that this group is the first to get something out on this topic. There are aspects of the study that are less than desirable like 1) lack of a positive control(s), 2) all field plots with D7 applications were also treated with herbicide, & 3) lack of caveat statements. Assuming treated plots controlled cheatgrass then I think your study helps lay to rest the notion that D7 needs to be applied with herbicide as it seemingly failed to establish and effects were likely 100% from the herbicide. Companies did some great marketing of off label use.

At the very least, the authors need some major revision to the discussion to critically report strengths and weaknesses of their study and how others should proceed. Readers will appreciate that the group sampled what was available and not necessarily what would have been ideal in hindsight.

Thank you for the suggestion. We have added to the discussion of our results and included caveat statements to provide transparency on the shortcomings of our experimental design. Please see lines 71-75, 98-101, and 119-121 of the manuscript.

Additional info-

Ann Kennedy did some early work with antibiotic resistant D7 & ACK55 strains. She used plate counts of soil dilutions to confirm short term D7 establishment (Johnson et al., 1993; Stubbs et al., 2014). Ann also reported longer term presence of ACK55 at field sites (Kennedy, 2018).

She wrote- "During a thaw in February following application in mid-fall, the bacterial strains were in soil at 5×10^8 cfu g⁻¹ soil. Two years after application, February bacterial populations in soil were at 6×10^7 cfu g⁻¹ soil. By the fourth year after application, the rifampicin-resistant populations in soil declined to levels of 3×10^5 cfu g⁻¹ soil, which was the lowest level that showed inhibition in the laboratory. Bacterial populations decreased faster in soils at sites with lower precipitation, although bacterial populations were not always different due to site moisture. Five to six years after application, the bacterial populations were below reliable detection levels in the field soil."

Additional comments-

L64 change "no" to "little". (& L124-6)

Note that Reinhart et al. (2019) performed Petri-plate, pot, and field experiments. The bioherbicide affected cheatgrass in Petri-plate but not in pots and field experiments thereby suggesting properties of regional soils (e.g. nutrients, texture, pH) may limit establishment of bioherbicide. Two pot experiments were also conducted that help to parse effects of resident microbes from other soil properties. For example, one experiment had pots filled with sterilized soils (main manuscript), two seed sources, etc. and another had varying soil types and soil inoculants (soil inoculant, soil inoculant+ACK55, ACK55; in supplement). These experiments were at ~optimal conditions (cool & wet) for bioherbicide, and they confirmed starting bacterial densities (so there was no question bacteria were originally alive and at a known density)...

Done. We had also added a reference to the Reinhart et al. paper on Line 60 (previous draft line 64).

L71 I'm sort of viewing your study as a trial and proof of concept. However, please include more details on experimental and sampling designs (e.g. plot #, samples per plot?). Or direct readers to the supplement for more detail.

Done. Given the length restrictions of the observation format, we refer readers to the supplementary information for details on experimental design. Please see the section headings "Soil collection and processing".

L71 Good treatments & well timed. Ann Kennedy would recommend applying D7 prior to rain/snow events so that the water would help carry the bacteria into the soil. Was this likely? How many plots? Not sure the experiment design was fully described even in the supplement (e.g. total number of plots, subplots?).

We have now indicated that treatments were applied in November to take advantage of increased soil moisture from winter snow. Please see line 74-75. Detailed experimental design and plot layout can be found in the supplementary materials. Specifically, this information can be found under the "Soil collection and processing" heading.

L74 Is this a good region for bacterial community composition? I assume yes but citations might be nice. **This is the most common region sequenced for bacterial community composition. Citations for the primers used are found in the supplementary information. Please see the section heading "DNA**

extractions and library prep”.

L80-2 Is this an informative region capable of differentiating species and isolates (for lack of a better word) and was there a positive control? Other evidence that this can A) detect *P. fluorescens* in general and B) differentiate D7 from other genotypes? Ann indicated the molecular tools (Ibekwe et al., 2010) were not able to detect D7 was that wrong?

Amplicon sequencing of the 16S rRNA gene is one of the most used molecular technologies and offers reliable and reproducible results for analyzing bacterial communities. We have read the paper by Ibekwe et al. (2010). The methods used (e.g., Random amplified polymorphic DNA (RAPD) are now outdated and provide very coarse grain information that does not address whether D7 is detectable by molecular tools, especially modern amplicon sequencing-based approaches. Instead, the authors determined whether the RAPD method could differentiate DNA fragments from mutated and non-mutated D7 strains. As the fragments are imaged on gels, there is no taxonomic information generated. Our amplicon approach does generate this information and is capable of assigning taxonomy at the species level in some cases. While this region of the 16S rRNA gene may not be able to capture strain-level differences, the lack of differences in abundance across members of Pseudomonadaceae between the treated plots and the controls provides support for our conclusion.

The taxonomic assignments generated in our bioinformatics pipeline were assigned using the SILVA database (SILVA). The SILVA db is the most widely cited and curated database for bacterial taxonomic assignments. This approach allowed us to search our taxonomic assignments for putative members of *P. fluorescens* across multiple levels of taxonomy to ensure all potential members of *P. fluorescens* were captured for downstream analyses. To ensure we had not missed possible taxonomic assignments, we then used the BLASTN database through NCBI using both the refseq_rna and nt (nucleotide collection) database, with and without the organism filter set to *Pseudomonas fluorescens*, and searched the top 100 potential hits for the species assignment “*fluorescens*” or strain “D7” at sequence similarity of 97% or higher. Our approach used modern molecular methods with a higher level of sensitivity than the methods used by Ibekwe et al. Additionally, we use two different taxonomic assignment approaches (e.g., SILVA db and BLASTN) to capture any potential members of *P. fluorescens*.

L88 please list actual P-values throughout.

The p-value on (old version) line 88 represents the results from several tests of different members of Pseudomonadaceae. All p-values were greater than 0.05. Indicating a single value would not be appropriate here. The supplementary results provide more detailed information on these tests and the abundance of these ASVs. Where we can use a single value we have (See Line 95 , e.g. $p = 0.19$). We have used the categories of i) $p < 0.05$, iii) $p < 0.01$, and iv) $p < 0.001$ throughout the manuscript. This allows readers to quickly gauge the level of significance.

L96 This could be a sign of two possible legacy effects: effect of nutrients in carrier solution (assuming residual growth media varied with concentration differences) or effect of D7 (though it seems unlikely since they were evidently more abundant 18 months later).

Agreed, and we lean toward the second explanation of a legacy effect. This is line with the findings of Malon et al. 2018 who showed a transient invader could leave lasting effects on community function and structure. We include this in our discussion on lines 106-116.

General comment- Microbial community descriptions may describe a large proportion of non-active soil microbes. Is it possible D7 was more prevalent during the winter months? I think this is unlikely for WY but it was isolated from soils in eastern Washington which likely has more favorable conditions for activity in winter (moist soils & not too cold) and possibly dramatically different soils (e.g. pH) than

WY... Ann did a lot of her monitoring of Ack55 in winter.

It is certainly *possible* that it would be more active in the winter months. However, the top layers of soils are likely frozen or have temperatures close to freezing. As you mentioned, the methods used to describe soil microbial communities often detect inactive microbes, So, if D7 did establish and grow, then it would be expected to be recovered in our sequence data.

L129 Comment- I suspect we need to have a better understanding of where the soil and abiotic conditions are suitable for D7 establishment relative to where D7 was locally adapted.

Agreed. We have added additional language in the closing section of the manuscript to indicate that this should be considered.

Discussion-

The discussion is a bit too long for such thin results. I would cut some of the speculative discussion content and add caveat content and future directions (or best practices). For example, it would have been nice to have samples collected pre-treatment, 1 wk post-treatment, & 18 months post-treatment (and in areas without herbicide). A positive control(s) would have been reassuring that the magic freeze-dried stuff was actually viable and D7. Having D7 applications with herbicide may have contributed to failure of D7 to establish. The isolate is thought to feed off of cheatgrass root exudates over winter. If you kill cheatgrass seedlings and few/none are overwintering then you likely starved D7. I suppose there is also the issue of DNA techniques describing numerous biota that are not active versus active. Many questions. **Thank you for your concerns. We have left the discussion largely intact as it incorporates a large body of literature from invasion biology theory and microbial invasions, both of which are important for interpreting and understanding our results. However, we included caveat statements to highlight the limitations of our study.**

Supplement-

L31 Is it possible that the herbicide confounded detecting D7? My impression from Ann is that the bacteria should proliferate with cheatgrass and feeds off of its root exudates (i.e. is closely associated with its roots but doesn't cause noticeable pathology of roots). If cheatgrass was suppressed by herbicide then that might hamstring detecting D7. To me, this herbicide + bioherbicide application scheme was created out of some business scheme and not biological processes. Ann gave me numerous documents related to instruction on use and none that I recall suggested applying D7 with herbicide. That would ignore the life cycle of D7.

While we agree that the herbicide application may have contributed to the failure to establish, land managers most commonly use this approach. In many studies (See Special Section of Rangeland Ecology and Management: "Special Section: Weed Suppressive Bacteria", specifically the editorial from Germino and Lazarus. DOI: 10.1016/j.rama.2020.02.007) no reduction of cheatgrass has been observed with only D7 application, so herbicide is required to reduce cheatgrass populations. We acknowledge that this is a limitation of our study but do not feel that it invalidates our result of not finding D7 in our sequence data.

L32 not sure about "synergistic effects" but realize vendor reps were pushing this idea at the time...

We have removed this statement as synergistic effects are not found in the Tekiela 2020 paper.

L33 Not seeing the citation for #1.

A bibliography is now included in the supplementary materials.

L57-8 You have a negative control. However, I wish that you had a positive control or something showing

that the bacteria were ever alive and capable of establishing. I have questions about whether the inoculant was ever viable, had D7 to begin with, whether something about the application killed D7.

While we agree with this sentiment, at this point, we are unable to include a positive control. We have indicated this shortcoming in the observation's discussion and suggest that future studies should include both positive and negative controls. Additionally, we added the following in the supplementary materials "No positive controls were included in the experimental design (e.g., plating of inoculant to ensure viability)."

L104 Have you done any trials to confirm that D7 is detectable in treated soils with other *Pseudomonas* &/or that the inoculant was ever viable and containing D7?

Using the molecular methods described in our observation and the same primer set, we have detected other members of *Pseudomonas*. This is true in this study as well.

References

Thank you for the references. We have read and incorporate where appropriate.

Ibekwe, A. M., Kennedy, A. C., & Stubbs, T. L. (2010). An assessment of environmental conditions for control of downy brome by *Pseudomonas fluorescens* D7. *International Journal of Environmental Technology and Management*, 12(1), 27-46.

Johnson, B. N., Kennedy, A. C., & Ogg, A. G. (1993). Suppression of downy brome growth by a rhizobacterium in controlled environments. *Soil Science Society of America Journal*, 57(1), 73-77. <https://doi.org/10.2136/sssaj1993.03615995005700010014x>

Kennedy, A. C. (2018). Selective soil bacteria to manage downy brome, jointed goatgrass, and medusahead and do no harm to other biota. *Biological Control*, 123, 18-27. <https://doi.org/https://doi.org/10.1016/j.biocontrol.2018.05.002>

Reinhart, K. O., Carlson, C. H., Feris, K. P., Germino, M. J., Jandreau, C. J., Lazarus, B. E., Mangold, J., Pellatz, D. W., Ramsey, P., Rinella, M. J., & Valliant, M. (2019). Weed-suppressive bacteria fails to control *Bromus tectorum* under field conditions. *Rangeland Ecology & Management*, accepted 7/25/2019.

Stubbs, T. L., Kennedy, A. C., & Skipper, H. D. (2014). Survival of a rifampicin-resistant *Pseudomonas fluorescens* strain in nine mollisols. *Applied and Environmental Soil Science*, 2014, 7, Article 306348. <https://doi.org/10.1155/2014/306348>

Re: Spectrum01771-23R2 (Failure to recover *Pseudomonas fluorescens* D7 supports claims of ineffectiveness as biocontrol agent of *Bromus tectorum*)

Dear Dr. Gordon Fritz Custer:

I am glad to inform you that your observation note has been accepted for publication in Microbiology Spectrum. The concerns and comments expressed by the two reviewers have been mostly attended and I acknowledge that the observation format precludes adding extensive discussion of the Results.

Your manuscript has been accepted, and I am forwarding it to the ASM production staff for publication. Your paper will first be checked to make sure all elements meet the technical requirements. ASM staff will contact you if anything needs to be revised before copyediting and production can begin. Otherwise, you will be notified when your proofs are ready to be viewed.

Sincerely,
Frédérique Reverchon
Editor
Microbiology Spectrum